# A Low-Protein High-Fat Diet Leads to Loss of Body Weight and White Adipose Tissue Weight via Enhancing Energy Expenditure in Mice

**DOI:** 10.3390/metabo11050301

**Published:** 2021-05-11

**Authors:** Yifeng Rang, Sihui Ma, Jiao Yang, Huan Liu, Katsuhiko Suzuki, Chunhong Liu

**Affiliations:** 1College of Food Science, South China Agricultural University, Guangzhou 510642, China; jackrang@stu.scau.edu.cn (Y.R.); yangjiao_626@163.com (J.Y.); liuhuanjzcd@126.com (H.L.); 2The Key Laboratory of Food Quality and Safety of Guangdong Province, Guangzhou 510642, China; 3Faculty of Sport Sciences, Waseda University, Tokorozawa 3591192, Japan; masihui@toki.waseda.jp; 4Japan Society for the Promotion of Sciences, Chiyoda-ku, Tokyo 1020083, Japan

**Keywords:** obesity, low-protein high-fat diet, fatty acid oxidation, energy expenditure

## Abstract

Obesity has become a worldwide health problem over the past three decades. During obesity, metabolic dysfunction of white adipose tissue (WAT) is a key factor increasing the risk of type 2 diabetes. A variety of diet approaches have been proposed for the prevention and treatment of obesity. The low-protein high-fat diet (LPHF) is a special kind of high-fat diet, characterized by the intake of a low amount of protein, while compared to typical high-fat diet, may induce weight loss and browning of WAT. Physical activity is another effective intervention to treat obesity by reducing WAT mass, inducing browning of WAT. In order to determine whether an LPHF, along with exercise enhanced body weight loss and body fat loss as well as the synergistic effect of an LPHF and exercise on energy expenditure in a mice model, we combined a 10-week LPHF with an 8-week forced treadmill training. Meanwhile, a traditional high-fat diet (HPHF) containing the same fat and relatively more protein was introduced as a comparison. In the current study, we further analyzed energy metabolism-related gene expression, plasma biomarkers, and related physiological changes. When comparing to HPHF, which induced a dramatic increase in body weight and WAT weight, the LPHF led to considerable loss of body weight and WAT, without muscle mass and strength decline, while it exhibited a risk of liver and pancreas damage. The mechanism underlying the LPHF-induced loss of body weight and WAT may be attributed to the synergistically upregulated expression of *Ucp1* in WAT and *Fgf21* in the liver, which may enhance energy expenditure. The 8-week training did not further enhance weight loss and increased plasma biomarkers of muscle damage when combined with LPHF. Furthermore, LPHF reduced the expression of fatty acid oxidation-related genes in adipose tissues, muscle tissues, and liver. Our results indicated that an LPHF has potential for obesity treatment, while the physiological condition should be monitored during application.

## 1. Introduction

The incidence of obesity has sharply increased in the last three decades, reaching two billion individuals over the world [1]. Obesity has become a global health problem among people of all ages [2]. It threatens the cardiovascular system and degrades life quality, ultimately decreasing life expectancy [3]. Body mass index (BMI) is routinely used to classify obesity. It is computed as body weight (kg) divided by the squared height (m^2^). BMI does not consider the body composition, especially body fat, which plays a crucial role in the risk assessment of metabolic disorders associated with obesity [2,4]. To accurately reflect the body composition, many novel indexes have been proposed for routine use in clinical practice, for example, waist-to-height ratio (WHtR), body adiposity index (BAI), and visceral adiposity index (VAI) [4,5]. Although obesity may be caused by a variety of environmental factors, diet plays a crucial role on adiposity [6]. Nowadays, many novel diets, for example, the high-fat “ketogenic” diet and very-low-carbohydrate “Atkins” diet are reported to have positive effects on the prevention and treatment of obesity [7,8,9].

A low-protein high-fat diet (LPHF) is a nutritional regimen characterized by consuming low amounts of protein, a moderate amount of carbohydrates, and high amounts of fat. It has been established that the majority of body fat is stored in adipose tissues, which mainly consist of white adipose tissue (WAT) and brown adipose tissue (BAT) in mammals [10]. During obesity, WAT tends to become severely dysfunctional, characterized by macrophage infiltration and fibrosis, inducing low-grade inflammation, systemic insulin resistance, and an increased risk of type 2 diabetes [11,12,13]. A recent study showed that LPHF induced weight loss in mice and raised the possibility that LPHF might have potential in body weight management [14]. However, the impacts of LPHF and LPHF combined with training on the metabolism of various tissues, especially WAT, remains unclear.

Fibroblast growth factor (FGF) 21 is regarded as an endocrine signal of protein restriction, expressed mainly in the liver [15,16]. In mice, liver *Fgf21* expression is upregulated within 24 h after protein restriction, leading to elevated plasma FGF21 concentrations [15]. FGF21 has been widely reported to cause weight loss by increasing energy expenditure [17]. It has been established that a high-fat diet plays an important role in energy expenditure such as fatty acid oxidation, carbohydrate utilization, and thermogenesis [18,19,20]. When consuming a low-carbohydrate high-fat ketogenic diet, an elevated plasma level of FGF21 is observed and, thus, contributes to weight loss [18,21]. Nevertheless, it remains unclear how FGF21 and energy expenditure will be regulated when LPHF and training are combined.

It has been established that endurance training contributes to body weight management and metabolic fitness [22], mainly involving several endocrine organs or tissues, for example, liver, adipose tissue, and muscle tissue [18]. Commonly, post-exercise protein intake is recommended, but a restricted protein diet may also elicit a positive impact [23], for example, alleviating the acidogenic effects of training [24]. Moreover, high protein intake is reported to increase cancer, diabetes and overall mortality [25]. Therefore, whether training in combination with LPHF could further promote weight loss and benefit energy metabolism is worth studying. In the current study, a 10-week LPHF feeding regimen with an 8-week treadmill training was designed, aimed at observing the changes in body weight as well as energy metabolism-related gene expression, plasma biomarkers, and related physiological changes. Meanwhile, a high-protein high-fat diet (HPHF) containing the same fat content and relatively more protein was introduced as a comparison.

## 2. Results

### 2.1. Change in Body Weight

As presented in Figure 1, after training began at Week 2, there was a significant difference between the bodyweight of the animals in the LPHF group and the HPHF group. From Week 4, the average body weight in LPHF groups was significantly lower than that in the Con groups. At the end of Week 8, the average body weight in the HPHF group was significantly higher than that in the Con. Training had no significant effect on Con groups, as well as the LPHF groups, while a significant difference was observed between HPHF and HPHF+T from Week 8.

### 2.2. Relative Tissue or Organ Weight

As presented in Table 1, the tissue/organ ratio of BAT (brown adipose tissue) weight was not significantly affected by LPHF, HPHF or training. Compared to LPHF and Con, the weight of WAT significantly increased in HPHF. The WAT weight in HPHF+T was significantly higher than that in Con+T. Diets showed significant impacts on SKM (skeletal muscle). There existed a significant difference in SKM weights between LPHF and HPHF. The liver weight significantly increased in LPHF, compared to the Con. Kidney weight was not significantly altered by diets. In Con groups and HPHF groups, training significantly decreased WAT weight, while it significantly increased SKM and kidney weight in the HPHF groups.

### 2.3. Grip Power and Maximal Exercise Capacity Test

Grip power was determined to evaluate the impacts of LPHF, HPHF, and training on muscle strength. As presented in Figure 2, training had a positive impact on grip power in Con groups. From Week 6, grip power was higher in the Con+T group than that in the Con groups. Training did not show a positive effect on grip power in both LPHF groups and HPHF groups. At Week 6, there was observed a trend in the increase in grip power between LPHF and LPHF+T as well as between HPHF and HPHF+T, but these trends disappeared at the end of the study. HPHF showed a positive effect on grip power. At Week 10, the grip power was higher in the HPHF group than that in the LPHF and Con groups. In contrast, LPHF did not affect grip power. Although grip power was lower in the LPHF group than in the Con group at Week 2, there was no obvious difference between the LPHF and Con groups from Week 6.

As presented in Figure 3, when the 10-week feeding finished, training showed an enhanced impact on the maximal exercise capacity in Con groups, LPHF groups, and HPHF groups. There were significant differences between the Con and Con+T groups, between the LPHF and LPHF+T groups as well as between the HPHF and HPHF+T groups. Compared to Week 2, the 8-week training enhanced the maximal exercise capacity in the Con+T, LPHF+T as well as the HPHF+T groups compared to their sedentary counterparts. Of note, the maximal exercise capacity was significantly higher in the LPHF+T group than that in the HPHF+T group. Trends in decreasing maximal exercise capacity were observed in the Con, LPHF, and HPHF group when comparing the results of Week 10 to the results of Week 2. Throughout the whole study, there was no significant differences at the same time point among the Con, LPHF, and HPHF groups.

Considering the considerable changes in body weight among the groups, the maximal workload was computed (Table 2). The maximal workload was significantly lower in LPHF than that in the Con and HPHF groups. Compared to the Con+T and LPHF+T groups, the maximal workload significantly increased in the HPHF+T group. Training significantly elevated maximal workload in the Con, LPHF, and HPHF groups. There was no significant difference between the Con+T and LPHF+T groups.

### 2.4. Analysis of Plasma Biochemical Indexes

In our study, the levels of albumin, BUN (blood urea nitrogen), creatinine, glucose, T-CHO (total cholesterol), L-CHO (low-density lipoprotein cholesterol), H-CHO (high-density lipoprotein cholesterol), NEFA (non-esterified fatty acid), TG (triglyceride), UA (uric acid), and BOHB (beta-hydroxybutyric acid) in the plasma were measured. As presented in Table 3, diets did not significantly affect plasma creatinine, NEFA, TG, UA, and BOHB. Compared to the Con and LPHF groups, the plasma glucose was significantly higher in the HPHF groups. No significant difference in plasma glucose was observed between the Con groups and LPHF groups. Both the LPHF and HPHF groups had significantly increased levels of T-CHO, L-CHO, and H-CHO in plasma, compared to the chow diet. Furthermore, the levels of plasma T-CHO, L-CHO, and H-CHO were higher in the HPHF groups than those in the LPHF groups. When comparing to the chow diet, the LPHF groups had significantly decreased plasma albumin and BUN. Training had no significant effect on these analyzed biochemical indexes. In addition, levels of plasma FGF21 (fibroblast growth factor 21) were also determined in our study. As presented in Figure 4, plasma FGF21 levels were significantly elevated in the LPHF groups, compared to the Con and HPHF groups. There were no significant difference between the Con and HPHF groups. Training had a significant impact on the LPHF groups. There was no significant difference between the Con and Con+T groups as well as between the HPHF and HPHF+T groups, while the plasma FGF21 levels significantly decreased in the LPHF+T group compared to the LPHF group.

In our research, amylase, AST (aspartate transaminase), CK (creatinine kinase), LDH (lactate dehydrogenase), and lipase were used as organ or muscle damage markers. As presented in Table 4, levels of amylase, AST, CK, and lipase in plasma were not significantly altered by training. Diets had a significant effect on plasma amylase, AST, LDH, and lipase. Compared to the Con groups, the levels of AST and lipase in plasma significantly increased in the LPHF groups. Plasma amylase levels were significantly elevated by HPHF when comparing to Con. LDH in plasma significantly increased in Con+T and LPHF+T compared to HPHF+T. A trend in increasing plasma levels of LDH was observed in the LPHF+T group compared to the Con+T group. Plasma LDH was increased by training in the Con groups, despite no significance being observed. Conversely, the levels of plasma LDH were significantly higher in the LPHF+T group than those in the LPHF groups.

### 2.5. Gene Expression in Tissues and Organs

The expression of relevant metabolism-associated genes in various tissues and organs were analyzed. As presented in Table 5, compared to chow diet, a HPHF diet significantly downregulated the expression of *Atgl* (fatty acid mobilization), *Klotho* (FGF21 receptor), and *Pgc1α* (browning of WAT), while it significantly upregulated *Leptin* (energy metabolism) in WAT. The expression levels of *Adiponectin* (anti-inflammation) were significantly lower in the HPHF group than that in the Con group. Of note, the expression levels of *F480* (inflammation) were significantly upregulated in the HPHF group compared to the Con and LPHF groups. The expression levels of *Ucp1* (thermogenesis) were significantly upregulated by LPHF when compared to the Con and HPHF groups. The *Ucp1* expression levels were significantly higher in the LPHF+T group than those in the LPHF group. Diets did not significantly affect *Cd36* (fatty acid transportation), *Il-6* (Inflammation), *Prdm16* (browning of WAT), *Pparγ* (browning of WAT), *Cidea* (thermogenesis), and *Pparα* (browning of WAT). Training significantly downregulated *F480* in the HPHF groups and did not significantly alter the other genes in WAT.

As presented in Table 6, the levels of *Ucp1* (thermogenesis) expression in BAT were significantly upregulated by HPHF, compared to the Con and LPHF groups. LPHF or training did not significantly affect *Ucp1* expression. Other genes related to thermogenesis (i.e., *Cidea* and *Prdm16*), inflammation (i.e., *Il6* and *Il10*), and mitochondrial respiration (*Cs*) were not significantly affected by diets or training.

As presented in Table 7, in the gastrocnemius muscle, a kind of fast-twitched glycolytic muscle, LPHF significantly reduced the expression levels of *Cox4* (mitochondrial respiration), compared to that in the chow diet. When comparing to chow diet, LPHF downregulated *Hadh* (fatty acid oxidation) and *Hbdh* (ketone body metabolism) and HPHF upregulated them. There existed a significant difference in *Hadh* and *Hbdh* expression between LPHF groups and HPHF groups. HPHF significantly increased *Mct1* (ketone body transportation) expression, compared to LPHF and chow diet. Training significantly downregulated *Sirt1* (mitochondrial respiration) in the Con groups, LPHF groups, and HPHF groups. Other genes related to gastrocnemius muscle ketone body metabolism (i.e., *Acat1* and *Oxct1*), fatty acid transportation (*Cd36*), mitochondrial respiration (i.e., *Cs*, *Cytochrome c*, and *Tfam*), glycogen synthase (*Gs*), glycogen phosphatase (*Gp*), and glucose metabolism (*Pparγ*) were not significantly affected by diets or training.

As presented in Table 8, in the soleus muscle, a kind of slow-twitched oxidative muscle, compared to the LPHF and chow diet, HPHF significantly upregulated the expression levels of *Acat1* (ketone body metabolism), *Cox4* (mitochondrial respiration), *Cpt1α* (fatty acid oxidation), *Hbdh* (ketone body metabolism), *Hk2* (glycolysis), *Mcad* (fatty acid oxidation), *Mct1* (ketone body transportation), *Il-10* (inflammation), and *Acc2* (fatty acid synthesis). The expression levels of *Pgc1α* (fatty acid mobilization) were significantly higher in the HPHF+T groups than those in the Con+T and LPHF+T groups. When comparing to Con, the LPHF groups significantly downregulated *Il-10*. Of note, compared to the chow diet, both LPHF and HPHF robustly increased *Pepck* (glycolysis) expression, despite no significance being observed. Training significantly upregulated the expression levels of *Cox4* in the Con groups, LPHF groups as well as the HPHF groups. Neither diets nor training significantly altered *Il-6* (fatty acid oxidation). No synergistic effect of diet and training was observed.

As presented in Table 9, all the significant changes in gene expression in liver derived from the alterations with diets, not with training. The LPHF diet significantly upregulated the expression levels of *Hmgcs2* (ketogenesis) and *Fgf21* (energy metabolism), and downregulated *Pparα* (fatty acid oxidation) and *Pgc1α* (fatty acid mobilization) in liver, compared to the chow diet. When comparing the HPHF group to the Con, the expression levels of *Atgl* (fatty acid mobilization), *Klotho* (FGF21 receptor), and *Pgc1α* were significantly downregulated. The expression levels of *Pparγ* (glucose metabolism) were significantly higher in the HPHF+T group than those in the Con+T group. Inflammation-related genes (*F480, Il-6,* and *Cxcl2*) were not significantly altered by diets. Training had no significant effect on these genes in liver.

## 3. Discussion

In general, the HPHF diet is used as a high-fat control to induce obesity in animal models [26,27,28]. Compared to its high-fat counterpart diet, the LPHF diet induced a unique metabolic status in various organs and tissues, characterized by increased thermogenesis and alleviated inflammation in WAT tissue and overexpression of *FGF21* in the liver. At the end of our study, compared to the chow diet, a 10-week HPHF diet caused a significant increase in body weight and WAT weight. In contrast, a 10-week LPHF diet led to considerable weight loss and reduced WAT weight. The effect of LPHF on lowering body weight is in accordance with a recent research [10]. The increase in the relative liver ratio in the LPHF group might be attributed to the body weight loss (Appendix A). Moreover, diets did not alter BAT weight. These results indicate that LPHF might be a potential approach for obese individuals to reduce body weight as well as WAT weight. It has been established that diets low in protein (3–8% energy as protein) often induce higher appetite [29]; thus, the food intake and corresponding energy intake in the LPHF groups were much higher than those in the HPHF groups (Appendix A). Therefore, we hypothesized that the weight loss caused by the LPHF diet might be attributed to the increased energy expenditure instead of appetite suppression [30,31].

In our study, an 8-week treadmill training was applied to examine whether training could benefit the management of body weight when an LPHF diet was applied. Considering that training in the daytime might bring greater health outcomes to animals, the treadmill training was conducted when the lights were on during our study [32]. Training is regarded as a weight-loss strategy in the treatment of obese individuals [33], while few studies have been performed in individuals with normal weight. A study reported that regular physical exercise in an apparently healthy elderly population caused no change in body weight or adiposity [34]. During the current study, an 8-week training reduced considerable WAT weight regardless of diets. Training also ameliorated the trend in body weight increase when consuming a HPHF diet and reduced body weight in the Con groups. Increased infiltration of macrophages was observed in the HPHF group, and training significantly reversed the effect. In the Con groups and HPHF groups, the increased kidney relative ratio might be due to the weight loss caused by training (Appendix A). However, training did not further enhance the weight loss caused by the LPHF diet. Moreover, in accordance with previous studies, the 8-week training increased plasma LDH (a marker of muscle damage) when compared to chow diet [35]. Training, to a higher extent, increased plasma LDH when LPHF was applied, indicative of elevated muscle damage. In contrast, HPHF along with training did not induce muscle damage in our study. Training was reported to stimulate rates of muscle protein synthesis [36,37]. Accordingly, protein deficiency in an LPHF diet may aggravate the muscle damage caused by training, and this result raised concern. Since we designed a relatively intensive training plan (5 days per week, treadmill speed was set at 22.5 m/min), further studies are encouraged to observe the synergistic effects of an LPHF diet and trainings of lower intensity.

Some studies have revealed that diets low in protein might cause the loss of muscle volume [38]. However, our results showed that LPHF diets did not alter the corresponding muscle mass. Muscle weight and muscle strength were also measured by a grip test, which was used to evaluate the explosive force. At the end of our study, the LPHF group did not have a reduced grip strength, and the HPHF group had robustly enhanced grip power. Explosive power was mainly dependent on fast-twist muscle, using glycometabolism as the energy source [39]. These results indicated that neither the LPHF diet nor the HPHF diet seemed to deactivate the glycometabolic capacity due to the fact of their moderate carbohydrate contents. A low-carbohydrate ketogenic diet was reported to compromise the explosive power in mice [18]. The increased body weight might result in enhanced grip power in HPHF. A recent review revealed that resistance training could increase muscle strength [40]. In our study, the 8-week training enhanced grip power in the Con groups. However, grip power was not increased by training when consuming an LPHF or HPHF diet. At the end of our study, mice in. the Con+T, HPHF, and HPHF+T groups had approximately the same grip power, which was higher than that in the LPHF+T group. Many studies have shown that extra protein is required to support muscle protein accretion and compensate catabolic loss of amino acids during strength training [41]. Protein deficiency in LPHF may be the reason why training failed to enhance grip power in the LPHF groups [42].

To evaluate the effect of LPHF and/or training on maximal exercise capacity, a graded treadmill exercise capacity test was employed. In our study, the maximal exercise capacity was reduced in the LPHF group. However, training robustly increased the maximal exercise capacity regardless of the diets, which is consistent with a previous study [43]. The maximal exercise capacity was higher in the HPHF+T group than that in Con+T and LPHF+T groups, which might be attributed to the increased muscle protein synthesis. It has been established that training and protein ingestion synergistically enhanced the process of muscle protein synthesis [37]. There was no significant difference between the Con+T and LPHF+T groups, which reveals that training might help to prevent the decline of endurance capacity when employing an LPHF diet. In order to explore how LPHF and/or training affected energy metabolism, including fat metabolism and carbohydrate utilization, we further analyzed the plasma biomarkers as well as gene expression of various tissues and organs.

Among analyzed biomarkers, LPHF reduced plasma albumin and BUN, which might be the consequence of the low protein content in LPHF. Both LPHF and HPHF increased the levels of T-CHO, L-CHO, and H-CHO in plasma, which was partly attributed to the high-fat contents. In accordance with a recent study conducted in Egypt [44], the plasma glucose levels were dramatically increased in the HPHF groups, compared to the Con and LPHF groups. Training lowered the plasma glucose in the HPHF groups, showing that regular training was favorable for blood glucose management in healthy individuals [45]. Plasma creatinine, NEFA, TG, and UA were not altered by diets or training. LPHF increased AST and lipase, indicators of hepatic damage and pancreatic damage, respectively. Previous studies reported that protein deficiency might lead to unfavorable changes in hepatic function via lowering hepatic antioxidative activity [46] as well as undesirable alterations of pancreatic function by altering secretory synergism in islets [47]. Amylase was increased by the HPHF diet. Amylase was used as another marker of pancreatic damage. A recent study reported that a HPHF diet induced pancreatic dysfunction via a JNK-related pathway [48]. Training did not affect amylase, AST, and lipase concentration/levels. CK, a marker of another muscle damage, was not altered by diets or training. To summarize, the LPHF diet increased the lipid pool and caused stress to liver and pancreas. Furthermore, the combination of a LPHF diet and training had no additive effects.

FGF21 is a hormone produced mainly in the liver by dietary protein restriction [49]. It was reported that FGF21 could enhance energy metabolism by upregulating uncoupling protein 1 (*Ucp1*) thus leading to weight loss [16,50,51]. *Ucp1* is known as a thermogenin and has crucial impacts on the balance of metabolism and energy, causing non-shivering thermogenesis by its mitochondrial uncoupling property [52]. In our study, an LPHF diet robustly increased plasma FGF21 and the expression level of *Fgf21* in liver. Meanwhile, the expression levels of *Ucp1* in WAT were significantly increased by the LPHF diet. In contrast, the HPHF diet did not increase plasma FGF21 concentrations and the expression levels of *Fgf21* in liver and *Ucp1* in WAT. Moreover, the expression levels of FGF21 receptor-related gene *Klotho* in WAT and liver were downregulated by HPHF. In general, BAT expresses higher levels of *Ucp1*, compared to WAT [11], while WAT was reported to have the capacity to dissipate energy as heat [53]. Considering these together with our results, an LPHF diet may induce loss of body weight and WAT weight through enhancing energy expenditure in WAT. In accordance with a previous study, the HPHF diet upregulated *Ucp1* in BAT [54], which might be due to the tissue-specific expression of *Ucp-1* based on the metabolism status [55]. It was reported that FGF21 might be required in energy expenditure caused by *Ucp1* [56]. A HPHF diet failed to cause weight loss, which might be due to the low levels of plasma FGF21 when consuming a HPHF diet. Future research is encouraged to directly analyze the change in heat production when consuming an LPHF diet. The current views regarding the impact of training on FGF21 are controversial [18,57]. According to our results, the 8-week training did not significantly affect FGF21 in the circulation as well as *Fgf21* expression in liver. Nonetheless, training reduced FGF21 levels in the LPHF groups, which might be attributed to the higher expression of liver FGF21 receptor-related gene *Klotho* in the LPHF+T group, compared to that in the LPHF group [58]. In addition, FGF21 was related with increased appetite in animals [59]. In accordance with the changes in FGF21, food intake decreased in the LPHF+T group, compared to the LPHF group (Appendix A). Considering that training was widely reported to increase appetite [60], we speculate that a synergistic effect between an LPHF diet and training may suppress appetite. Furthermore, the expression levels of *Ucp1* were enhanced by training in LPHF groups, while no additive effects were observed when combining diets with training.

In WAT, the HPHF diet downregulated the fatty acid mobilization-related gene *Atgl* and browning of the WAT-related gene *Pgc1α*. These results indicate that the HPHF diet reduced fat consumption in WAT, which was consistent with the finding that the WAT relative ratio was increased by the HPHF diet. Moreover, the expression levels of *Leptin* were upregulated by HPHF. *Leptin* plays an important role in energy homeostasis [61], and elevated leptin levels in obese individuals are interpreted as leptin resistance [62]. The upregulated *Leptin* might be due to the increased weight in HPHF groups. In muscle, LPHF reduced fat consumption by downregulating *Hadh* and *Hbdh* in gastrocnemius muscle. *Cox4* expression was also downregulated by an LPHF diet in gastrocnemius muscle, indicative of the decreased capacity of fatty acid oxidation. In contrast, despite upregulating fatty acid synthesis-related gene *Acc2* in soleus muscle, a HPHF diet enhanced fat consumption by upregulating *Hadh*, *Hbdh* and *Mct1* in gastrocnemius muscle, as well as *Acat1*, *Cpt1α*, *Hbdh*, *Mcad*, and *Mct1* in soleus muscle. The expression levels of soleus muscle *Pgc1α* were higher in the HPHF+T group than those in Con+T. The HPHF diet also upregulated the mitochondria respiration-related genes, *Cox4* and *Tfam*, in soleus muscle, indicative of an increase of fatty acid oxidation capacity. The increased fat consumption might contribute to the decreased skeletal muscle relative ratio in the HPHF groups. Fat is an important substrate for muscle contraction, both at rest and during training [63]. Our results indicated that protein contents in diets had a crucial impact on fat metabolism in muscle. Further research is needed to explore the specific mechanism. Training was reported to alter gene expression in skeletal muscle [64,65]. In our study, training downregulated *Sirt1* (mitochondrial respiration) in gastrocnemius muscle and upregulated *Cox4* (mitochondrial respiration) in soleus muscle. No additive effects were observed when combining diets with training. In liver, an LPHF diet reduced fatty acid oxidation by downregulating *Pparα* and *Pgc1α*. A HPHF diet also reduced fatty acid oxidation by downregulating *Atgl* and *Pgc1α* in liver. The decreased capacity of fatty acid oxidation in liver might contribute to the elevated levels of T-CHO, L-CHO, and H-CHO in plasma when consuming an LPHF or HPHF diet. Training did not affect the fatty acid oxidation-related genes in liver. To summarize, an LPHF diet reduced the capacity of fat metabolism in WAT, muscle, and liver. Furthermore, the combination of diets and training had no additive effects.

Despite the high fat content (60.0% energy as fat), neither an LPHF nor HPHF diet increased plasma BOHB, which may be due to the moderate carbohydrate content in LPHF (35.1% energy as carbohydrate) and HPHF (25.8% energy as carbohydrate) diets, as carbohydrates have been reported to be anti-ketogenic [66]. An LPHF diet elevated the expression levels of *Hmgcs2* (ketogenesis) in liver; *Hmgcs2* plays a critical role in ketogenesis, which is mainly conducted in the mitochondria of hepatocytes. Ketogenesis is involved in various reactions that lead to the production of ketone bodies. BOHB is the major ketone body in the plasma. Ketogenesis requires efficient mitochondrial β-oxidation of fatty acids [67]. Despite the elevated *Hmgcs2* expression in liver, LPHF did not increase the plasma ketone body due to the reduced fatty acid oxidation.

It was reported that glucose metabolism was necessary for adequate metabolic performance when consuming a high-fat diet [68]. In soleus muscle, a HPHF diet increased glucose metabolism capacity by upregulating *Hk2* and *Pparγ*. Despite no significance being observed, both the LPHF and HPHF diets enhanced glycolysis-related gene *Pepck* expression in soleus muscle. In liver, the expression levels of glucose metabolism-related gene *Pparγ* were higher in HPHF+T than those in Con+T. The increased glucose metabolism capacity caused by the LPHF or HPHF diets might be due to the high-fat contents in both diets. In soleus muscle, the expression levels of inflammation-related gene *Il-10* were lower in LPHF than those in the Con group. In contrast, when comparing the HPHF group to the Con group, the anti-inflammation related gene *Adiponectin* was downregulated and the inflammation-related gene *F480* was upregulated in WAT. A recent review showed that diets with more dairy proteins have neutral to beneficial impacts on inflammation [69]. However, when considering the similar high-fat contents in both diets, our results showed that LPHF reduced inflammation in soleus muscle, while HPHF increased inflammation in WAT.

The limitation of our current study was that only the changes of gene expression levels in various tissues and organs were analyzed, while there lacked an alternative assay which was more directly linked to energy expenditure, for example, activity changes in certain key enzymes. Further studies are encouraged to provide direct evidences on the effects of an LPHF diet on energy expenditure, applying a different evaluating model. In addition, our study determined the common genes related with energy expenditure. By comparing with the HPHF group, our results revealed several common genes contributing to the energy expenditure when consuming an LPHF diet, for example, *Ucp1* and *Fgf21*. However, in contrast to other studies, most of these common genes were not upregulated in the LPHF group, although energy expenditure was dramatically enhanced. Therefore, other energy expenditure-related genes, such as G protein-coupled receptor related-genes, might be involved in the special energy metabolism status caused by an LPHF diet [70]. Future research is encouraged to study the other genes related with energy expenditure. Another limitation was the lack of an in-depth examination of the tissues or organs including the degree of damage and the fat contents. To make clear the mechanisms and prevent the side-effects when consuming an LPHF diet, future studies are encouraged to further analyze the tissues and organs. Furthermore, although our results revealed that increased energy expenditure might contribute to body weight loss when applying an LPHF diet, there might be other mechanisms, such as steatorrhea, due to the lack of fat content analysis in the stool during our current study.

The main finding of the current study was that an LPHF diet caused a considerable loss of body weight and WAT weight without muscle mass and strength decline. However, training did not further enhance the weight loss and increased muscle damage when consuming LPHF. The mechanism underlying an LPHF diet-induced loss of body weight and WAT weight may be attributed to the synergistically upregulated expression of *Ucp1* in WAT and *Fgf21* in liver, which enhanced energy expenditure. In our study, LPHF caused a considerable loss of body weight and WAT weight in mice with normal weight; thus, an LPHF diet is expected to have a positive effect on the management of obesity.

## 4. Materials and Methods

### 4.1. Mice Maintenance and Diets

Male C57BL/6J mice (*n* = 48) at 8 weeks of age were purchased from Takasugi Experimental Animals Supply (Kasukabe, Japan) and acclimated to the environment for one week. Four mice were housed in one cage with a specification of 27 × 17 × 13 cm. The environment was controlled with a 12 h light–dark cycle (lights were available at 08:00–20:00). The experiment was approved by the Academic Research Ethical Review Committee of Waseda University (10K001) and followed the Guiding Principles for the Care and Use of Animals. All mice were randomly divided into six groups: chow diet (Con), involving sedentary behavior (*n* = 8), chow diet plus training (Con+T, *n* = 8; T is the abbreviation for training), an LPHF involving sedentary behavior (*n* = 8), an LPHF plus training (LPHF+T, *n* = 8), a HPHF involving sedentary behavior (*n* = 8), and a HPHF plus training (HPHF+T, *n* = 8). The nutritional information (in energy) was presented in Table 10. All the diets were purchased from Trophic (TROPHIC Animal Feed High-Tech Co., Ltd., Jiangsu, China). Mice were kept on an ad libitum chow, LPHF, or HPHF, for 10 weeks, starting at the age of 9 weeks. Of note, our current study was conducted along with a published research focusing on a low-carbohydrate ketogenic diet, sharing the same results from the groups of Con and Con+T [18].

### 4.2. Training Procedures

Every training day, all mice in three training groups completed one hour of forced treadmill running at 18:00–20:00. Training was performed the first 5 days per week, using a treadmill (Natsume, Tokyo, Japan) with a five-grade slope. The speed of running was 22.5 m/min. The training started after 2 weeks of feeding chow, LPHF, or HPHF diets (at 11 weeks of age) and lasted 8 weeks.

### 4.3. Forelimb Grip Strength Test

We applied a grip-strength meter (GPM-100; Melquest, Toyama, Japan) to determine forelimb grip strength, according to a previously published method [18]. Grip power was determined at three time points: first at the beginning (Week 2), second at the mid-term (Week 6), and third at the end (Week 10). A force gauge was vertically fixed to a metal stand to immobilize the system. A mouse was allowed to grasp the bar mounted on the force gauge. When the mouse grasped the bar, a digital force transducer recorded the peak pull force in grams. The gauge was reset to 0 g after stabilization, and then the inspector slowly pulled back the mouse via its tail. Once the mouse released its two forepaws from the bar, tension was recorded on the gauge. Trials in which only one forepaw was used and the mouse turned when pulling its tail or released the bar without resistance were excluded. An operation at a sufficiently slow and constant speed was conducted to help mice to establish resistance against it. For each test, three consecutive measurements were performed at 30 s intervals.

### 4.4. Maximal Workload

A treadmill was used to evaluate maximal exercise capacity, according to a previously described method [18]. After familiarizing with running on a motorized rodent treadmill for one week before training began, all mice completed two graded exercise performance tests at two time points: 3 days before training and 3 days after training. Tests began at 9 m/min for 9 min, then increased to 10 m/min by 2.5 m/min every 3 min. The angle (θ) of the treadmill started at 0, raising to a maximal incline of 15 by 5 every 9 min. Exhaustion was defined as the status when mice were not able to continue regular running despite repeatedly tapping on the back. Running time (min) was recorded when it came to the status of exhaustion. The mouse was then returned into its home cage. Exercise time (min) and performed work (kg·m) were used to evaluate the exercise capacity. The performed work was equal to the product of body weight (kg), running distance (m), and sin θ, where θ ranged from 0 to 15.

### 4.5. Sampling and Plasma Biochemical Assessment

Sampling was conducted two days after the final training. Mice were fasted overnight for 12 h and then sacrificed under isoflurane-induced anesthesia [71]. Blood samples collected from the abdominal aorta with heparin were centrifugated (1500× *g*, 10 min, 4 °C) to obtain plasma. Tissues and organs were instantly excised and stored in liquid nitrogen. Plasma was obtained by centrifugating the blood samples at 1500× *g* for 10 min at 4 °C. All the samples were maintained at –80 °C until assessment. The tissues and organs were weighed, including the whole liver, WAT, BAT, kidney, and a muscle fascicle in the left hind leg, containing gastrocnemius, soleus, as well as plantaris muscles. Plasma levels of albumin, blood urea nitrogen (BUN), creatinine, glucose, total cholesterol (T-CHO); low-density lipoprotein cholesterol (L-CHO); high-density lipoprotein cholesterol (H-CHO); non-esterified fatty acid (NEFA); triglyceride (TG); uric acid (UA), amylase, aspartate transaminase (AST); creatinine kinase (CK); lactate dehydrogenase (LDH), and lipase were determined by Koutou-Biken Co. (Tsukuba, Japan). The concentrations of beta-hydroxybutyric acid (BOHB) were determined with a commercial assay kit (Cayman, MI, USA). Plasma fibroblast growth factor 21 (FGF21) levels were determined with Mouse FGF21 ELISA kit (R&D Systems, Minneapolis, MN, USA).

### 4.6. Real-Time PCR

Total RNA in the liver was extracted using TRIzol reagent, total RNA in the gastrocnemius muscle and the soleus muscle were extracted by the RNeasy Fibrous Mini Kit, and total RNA in the white fat tissue and the brown fat tissue were extracted by the RNeasy Lipid Tissue Mini Kit (Qiagen, Valencia, CA, USA). The purity and concentration of the total RNA were assessed by the NanoDrop system (NanoDrop Technologies, Wilmington, DE, USA), and the total RNA was reverse-transcribed by the High Capacity cDNA Reverse Transcription Kit (Applied Biosystems, Foster City, CA, USA). PCR was conducted by the Fast 7500 Real-Time PCR system (Applied Biosystems) with the Fast SYBR^®^ Green PCR Master Mix (Applied Biosystem). The thermal cycle procedure began with denaturation at 95 °C for 10 min, followed by 40 cycles at 95 °C for 3 s and annealing at 60 °C for 15 s. In the current study, 18s rRNA was applied as the housekeeping gene, and the ∆∆CT method was used to quantify target gene expression. All data are shown relative to their expression as fold change based on the results in Con. Primers used in the PCR are presented in Table 11.

### 4.7. Statistical Analysis

Test data in the tables and figures are shown as the mean ± standard deviation (SD). Two-way analysis of variance (ANOVA) and Turkey’s post hoc test were applied for statistical analysis using GraphPad Prism 8.0 (GraphPad, Ltd., LaJolla, CA, USA). Statistical significance was set at *p* < 0.05. ANOVA was used to determine the main impacts of diet and/or training. When ANOVA showed a significant impact from the interaction of diet and training, Tukey’s post hoc test was then conducted to measure the significance of differences between means.

## 5. Conclusions

In the current study, we applied a 10-week LPHF diet along with an 8-week treadmill training to investigate whether an LPHF diet could cause weight loss in mice with normal weight. The current results revealed that an LPHF diet caused considerable loss of body weight and WAT weight, which might be attributed to the synergistically upregulated expression of *Ucp1* in WAT and *Fgf21* in liver. Muscle mass and strength were not reduced by the LPHF diet. Despite enhancing the WAT weight loss, training did not further enhance body weight loss and increased muscle damage when consuming an LPHF diet. Therefore, an LPHF diet may be applied to reduce body weight, while its mechanisms and side effects on tissues/organs remain to be further explored.

## Figures and Tables

**Figure 1 metabolites-11-00301-f001:**
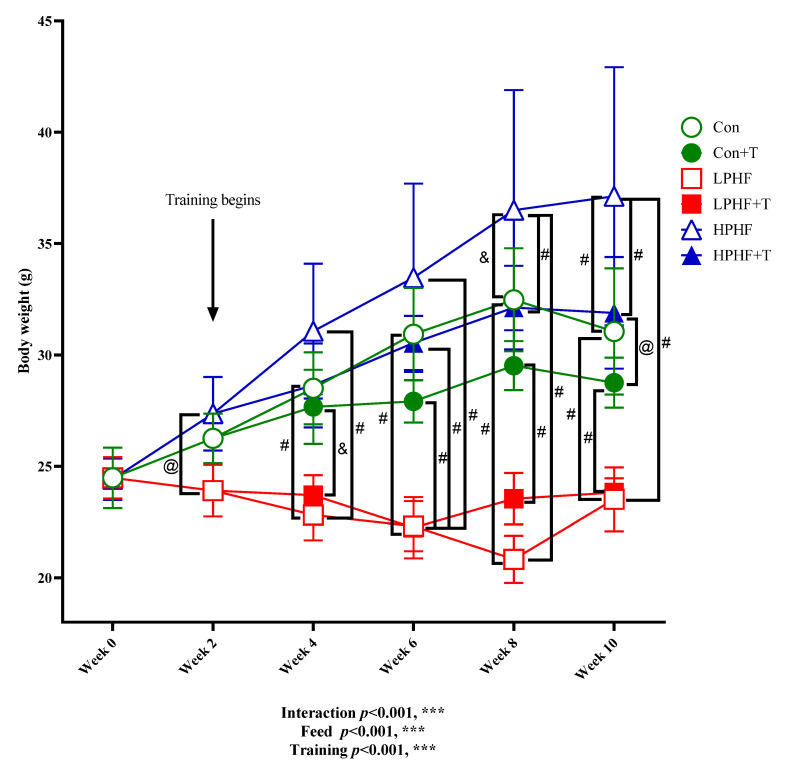
Changes in body weight throughout the experiment. Con, Con+T, LPHF, LPHF+T, HPHF, and HPHF+T represent chow diet, chow diet plus training, low-protein high-fat diet, low-protein high-fat diet plus training, high-protein high-fat diet, and high-protein high-fat diet plus training, respectively. Feed, training, and interaction indicate a dominant effect of diet, treadmill exercise, and the interactive effect between feed and training, respectively. *** *p* < 0.001. @, *p* < 0.05; &, *p* < 0.01; #, *p* < 0.001. Data are presented as the mean ± SD. *n* = 8 for each group. Significance of differences between means was determined by Tukey’s post hoc test when ANOVA revealed a significant effect from the interaction of diet and training.

**Figure 2 metabolites-11-00301-f002:**
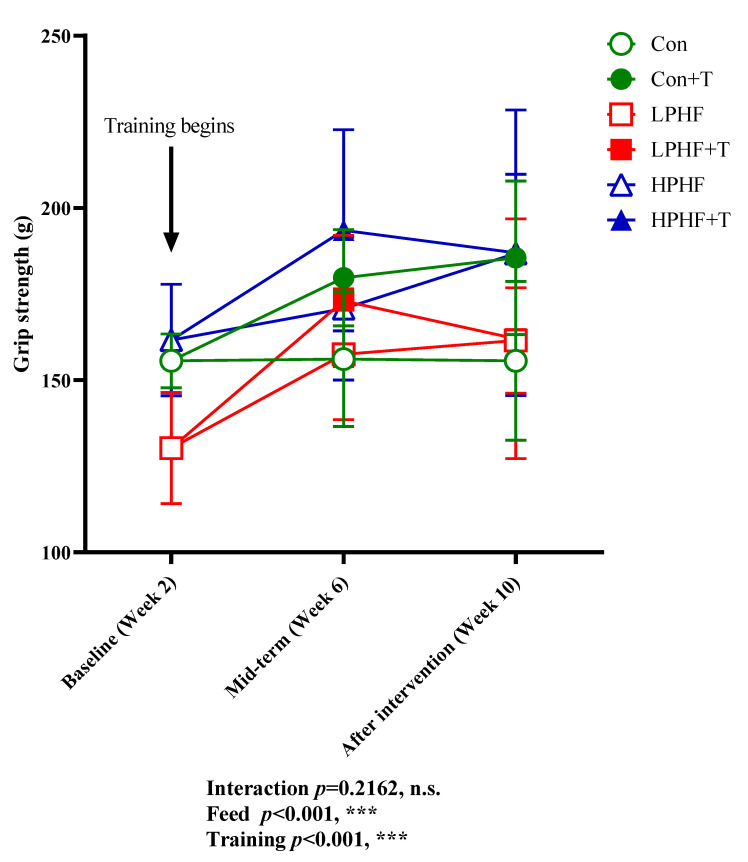
Forelimb grip power. Con, Con+T, LPHF, LPHF+T, HPHF, and HPHF+T represent chow diet, chow diet plus training, low-protein high-fat diet, low-protein high-fat diet plus training, high-protein high-fat diet, and high-protein high-fat diet plus training, respectively. Feed, training, and interaction indicate the dominant effect of diet, treadmill exercise, and the interactive effect between feed and training, respectively. n.s., no significance was observed. *** *p* < 0.001. Data are presented as the mean ± SD. *n* = 8 for each group.

**Figure 3 metabolites-11-00301-f003:**
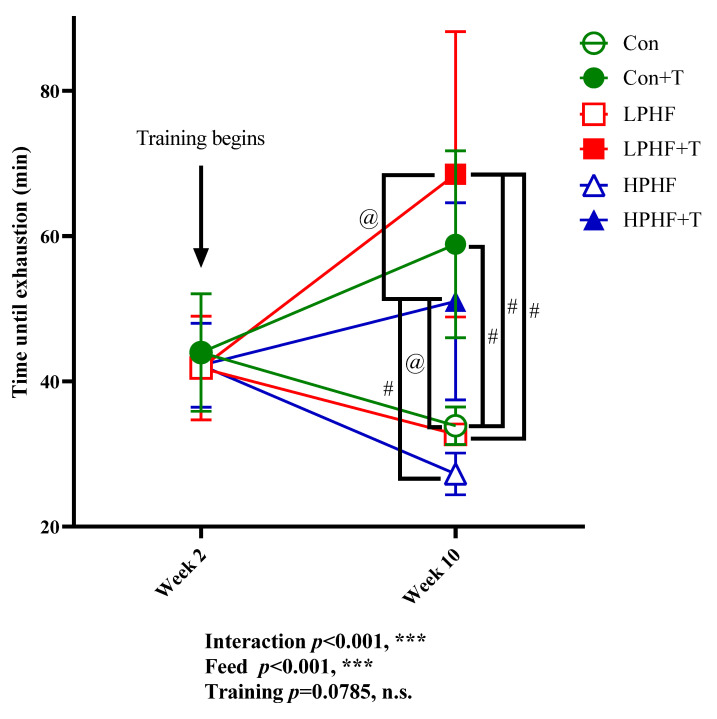
Maximal exercise capacity. Con, Con+T, LPHF, LPHF+T, HPHF, and HPHF+T represent chow diet, chow diet plus training, low-protein high-fat diet, low-protein high-fat diet plus training, high-protein high-fat diet, and high-protein high-fat diet plus training, respectively. Feed, training, and interaction indicate a dominant effect of diet, treadmill exercise, and the interactive effect between feed and training, respectively. n.s., no significance was observed. *** *p* < 0.001. @, *p* < 0.05; #, *p* < 0.001. Data are presented as the mean ± SD. *n* = 8 for each group. Significance of differences between means was determined by Tukey’s post hoc test when ANOVA revealed a significant effect from the interaction of diet and training.

**Figure 4 metabolites-11-00301-f004:**
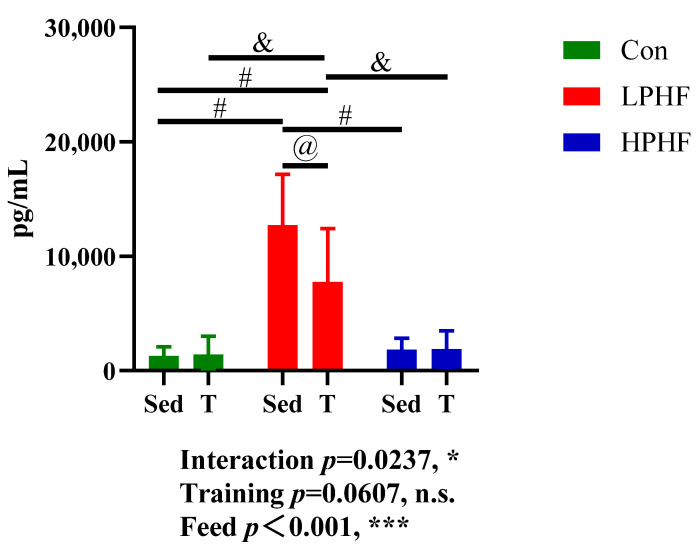
Plasma FGF21 concentration. FGF21, fibroblast growth factor 21. Con, LPHF, and HPHF represent chow diet, low-protein high-fat diet, and high-protein high-fat diet, respectively. Sed, sedentary behavior; T, training. Feed, training, and interaction indicate a dominant effect of diet, treadmill exercise, and the interactive effect between feed and training, respectively. n.s., no significance was observed. * *p* < 0.05 and *** *p* < 0.001. @, *p* < 0.05; &, *p* < 0.01; #, *p* < 0.001. Data are presented as the mean ± SD. *n* = 8 for each group. Significance in the differences between means was determined by Tukey’s post hoc test when ANOVA revealed a significant effect from interaction of diet and training.

**Table 1 metabolites-11-00301-t001:** Effects of LPHF, HPHF, and/or training on the relative ratio between tissue/organ weight and body weight.

Tissue/Organ	Con Groups		LPHF Groups		HPHF Groups		Significance		
	Con	Con+T	LPHF	LPHF+T	HPHF	HPHF+T	Training	Feed	Interaction
WAT, %	2.87 ± 0.79 ^be^	1.50 ± 0.61 ^aef^	2.25 ± 0.33 ^e^	1.89 ± 0.46 ^e^	5.09 ± 1.43 ^abcdf^	3.10 ± 0.92 ^be^	***	***	*
BAT, %	0.34 ± 0.07	0.33 ± 0.09	0.38 ± 0.09	0.34 ± 0.10	0.35 ± 0.07	0.27 ± 0.07	n.s.	n.s.	n.s.
SKM, %	0.58 ± 0.05	0.61 ± 0.03	0.60 ± 0.04	0.61 ± 0.06	0.50 ± 0.05	0.56 ± 0.05	**	***	n.s.
Liver, %	3.57 ± 0.33	3.82 ± 0.31	4.19 ± 0.57	4.10 ± 0.21	3.49 ± 0.40	3.76 ± 0.25	n.s.	***	n.s.
Kidney, %	1.04 ± 0.11	1.14 ± 0.12 ^e^	1.00 ± 0.06	1.03 ± 0.05	0.92 ± 0.11 ^bf^	1.13 ± 0.11 ^e^	***	n.s.	*

WAT, BAT, and SKM represent white adipose tissue, brown adipose tissue, and skeletal muscle, respectively. Con, Con+T, LPHF, LPHF+T, HPHF, and HPHF+T represent chow diet, chow diet plus training, low-protein high-fat diet, low-protein high-fat diet plus training, high-protein high-fat diet, and high-protein high-fat diet plus training, respectively. Feed, training, and interaction indicate a dominant effect of diet, treadmill exercise, and the interactive effect between feed and training, respectively. n.s., no significance was observed. * *p* < 0.05, ** *p* < 0.01, and *** *p* < 0.001. Data are presented as the mean ± SD. *n* = 8 for each group. Significance of differences between means was determined by Tukey’s post hoc test when ANOVA revealed a significant effect from interaction of diet and training. ^a–f^ Significantly different from Con, Con+T, LPHF, LPHF+T, HPHF, and HPHF+T, respectively.

**Table 2 metabolites-11-00301-t002:** Effects of LPHF, HPHF and/or training on maximal exercise workload after training intervention.

Index	Con Groups		LPHF Groups		HPHF Groups		Significance		
	Con	Con+T	LPHF	LPHF+T	HPHF	HPHF+T	Training	Feed	Interaction
MW (kg·m)	19,641 ± 1345 ^bcdf^	41,378 ± 1269 ^acef^	12,534 ± 884 ^abdef^	41,211 ± 2331 ^acef^	16,437 ± 2510 ^bcdf^	47,668 ± 3931 ^abcde^	***	***	***

MW, Maximal workload, determined at the end of Week 10. Maximal workload was defined as total climbed vertical distance × weight. Con, Con+T, LPHF, LPHF+T, HPHF, and HPHF+T represent chow diet, chow diet plus training, low-protein high-fat diet, low-protein high-fat diet plus training, high-protein high-fat diet, and high-protein high-fat diet plus training, respectively. Feed, training, and interaction indicate a dominant effect of diet, treadmill exercise, and the interactive effect between feed and training, respectively. n.s., no significance was observed. *** *p* < 0.001. Data are presented as the mean ± SD. n = 8 for each group. Significance of differences between means was determined by Tukey’s post hoc test when ANOVA revealed a significant effect from interaction of diet and training. ^a–f^, significantly different from Con, Con+T, LPHF, LPHF+T, HPHF, and HPHF+T, respectively.

**Table 3 metabolites-11-00301-t003:** Effects of LPHF, HPHF and/or training on plasma biochemical markers.

Index	Con Groups		LPHF Groups		HPHF Groups		Significance		
	Con	Con+T	LPHF	LPHF+T	HPHF	HPHF+T	Training	Feed	Interaction
Albumin, mg/dL	2.55 ± 0.23	2.44 ± 0.19	2.21 ± 0.16	2.29 ± 0.16	2.51 ± 0.16	2.33 ± 0.14	n.s.	***	n.s.
BUN, mg/dL	23.44 ± 5.02	25.09 ± 3.50	16.61 ± 2.41	17.48 ± 5.35	21.30 ± 4.11	21.26 ± 3.06	n.s.	***	n.s.
Creatinine, mg/dL	0.11 ± 0.03	0.12 ± 0.02	0.11 ± 0.04	0.11 ± 0.05	0.09 ± 0.02	0.09 ± 0.02	n.s.	n.s.	n.s.
Glucose, mg/dL	201.38 ± 58.67 ^e^	212.25 ± 33.78 ^e^	179.63 ± 37.35 ^e^	195.00 ± 39.70 ^e^	292.88 ± 45.75 ^abcdf^	216.75 ± 46.08 ^e^	n.s.	***	**
T-CHO, mg/dL	100.88 ± 10.13	93.38 ± 8.40	110.63 ± 23.70	113.25 ± 7.65	133.88 ± 22.56	116.25 ± 13.58	n.s.	***	n.s.
L-CHO, mg/dL	13.88 ± 3.56	15.00 ± 4.81	19.88 ± 8.17	19.13 ± 5.54	21.00 ± 7.35	20.25 ± 3.85	n.s.	**	n.s.
H-CHO, mg/dL	76.13 ± 8.17	69.75 ± 6.94	82.13 ± 19.11	85.88 ± 6.98	103.50 ± 16.35	88.88 ± 13.41	n.s.	***	n.s.
NEFA, µEq/L	2.85 ± 0.31	2.63 ± 0.20	2.45 ± 0.48	2.63 ± 0.18	2.75 ± 0.34	2.64 ± 0.44	n.s.	n.s.	n.s.
TG, mg/dL	20.63 ± 7.25	16.13 ± 7.85	12.38 ± 7.76	14.25 ± 6.76	13.71 ± 3.82	16.88 ± 7.16	n.s.	n.s.	n.s.
UA, mg/dL	1.39 ± 0.51	1.39 ± 0.51	1.46 ± 0.65	1.20 ± 0.32	1.13 ± 0.68	1.09 ± 0.45	n.s.	n.s.	n.s.
BOHB, mmol/L	0.40 ± 0.43	0.31 ± 0.36	0.39 ± 0.11	0.44 ± 0.10	0.33 ± 0.14	0.29 ± 0.13	n.s.	n.s.	n.s.

BUN, T-CHO, L-CHO, H-CHO, NEFA, TG, UA, and BOHB represent blood urea nitrogen, total cholesterol, low-density lipoprotein cholesterol, high-density lipoprotein cholesterol, non-esterified fatty acid, triglyceride, uric acid, and beta-hydroxybutyric acid, respectively. Con, Con+T, LPHF, LPHF+T, HPHF, and HPHF+T represent chow diet, chow diet plus training, low-protein high-fat diet, low-protein high-fat diet plus training, high-protein high-fat diet, and high-protein high-fat diet plus training, respectively. Feed, training, and interaction indicate a dominant effect of diet, treadmill exercise, and the interactive effect between feed and training, respectively. n.s., no significance was observed. ** *p* < 0.01 and *** *p* < 0.001. Data are showed as the mean ± standard deviation. *n* = 8 for each group. Significance of differences between means was determined by Tukey’s post hoc test when ANOVA revealed a significant effect from the interaction of diet and training. ^a–f^ Significantly different from the Con, Con+T, LPHF, LPHF+T, HPHF, and HPHF+T groups, respectively.

**Table 4 metabolites-11-00301-t004:** Effects of LPHF, HPHF, and/or training on plasma muscle/organ damage indicators.

Index	Con Groups		LPHF Groups		HPHF Groups		Significance		
	Con	Con+T	LPHF	LPHF+T	HPHF	HPHF+T	Training	Feed	Interaction
Amylase, IU/L	1756 ± 252	1623 ± 220	1635 ± 174	1761 ± 332	1989 ± 216	1833 ± 280	n.s.	*	n.s.
AST, IU/L	81 ± 47	69 ± 48	108 ± 64	132 ± 106	76 ± 40	66 ± 24	n.s.	*	n.s.
CK, IU/L	155 ± 130	136 ± 204	330 ± 307	231 ± 295	127 ± 96	113 ± 81	n.s.	n.s.	n.s.
LDH, IU/L	421 ± 290 ^d^	881 ± 299 ^f^	394 ± 99 ^d^	1190 ± 615 ^acef^	407 ± 277 ^d^	390 ± 202 ^bd^	**	***	**
Lipase, IU/L	40 ± 7	46 ± 6	64 ± 11	63 ± 7	47 ± 7	58 ± 21	n.s.	***	n.s.

AST, CK, and LDH represent aspartate transaminase, creatinine kinase, and lactate dehydrogenase, respectively. Con, Con+T, LPHF, LPHF+T, HPHF, and HPHF+T represent chow diet, chow diet plus training, low-protein high-fat diet, low-protein high-fat diet plus training, high-protein high-fat diet, and high-protein high-fat diet plus training, respectively. Feed, training, and interaction indicate a dominant effect of diet, treadmill exercise, and the interactive effect between feed and training, respectively. n.s., no significance was observed. * *p* < 0.05, ** *p* < 0.01, and *** *p* < 0.001. Data are shown as the mean ± standard deviation (SD). *n* = 8 for each group. Significance in the differences between means was determined by Tukey’s post hoc test when ANOVA revealed a significant effect from the interaction of diet and training. ^a–f^ significantly different from the Con, Con+T, LPHF, LPHF+T, HPHF, and HPHF+T groups, respectively.

**Table 5 metabolites-11-00301-t005:** Effects of LPHF, HPHF, and/or training on WAT gene expression.

Gene	Con Groups		LPHF Groups		HPHF Groups		Significance		
	Con	Con+T	LPHF	LPHF+T	HPHF	HPHF+T	Training	Feed	Interaction
*Adiponectin* (Anti-inflammation)	1.00 ± 0.46	1.01 ± 0.29	0.79 ± 0.32	0.69 ± 0.42	0.43 ± 0.13	0.79 ± 0.37	n.s.	*	n.s.
*Atgl* (Fatty acid mobilization)	1.00 ± 0.40	0.95 ± 0.43	0.77 ± 0.43	0.31 ± 0.18	0.35 ± 0.17	0.27 ± 0.10	n.s.	***	n.s.
*Cd36* (Fatty acid transportation)	1.00 ± 0.36	0.90 ± 0.29	0.87 ± 0.28	0.72 ± 0.38	0.62 ± 0.27	0.71 ± 0.37	n.s.	n.s.	n.s.
*F480* (Inflammation)	1.00 ± 0.30 ^e^	0.91 ± 0.20 ^e^	0.96 ± 0.13 ^e^	0.90 ± 0.61 ^e^	3.91 ± 1.84 ^abcdf^	0.99 ± 0.36 ^e^	***	***	***
*Il-6* (Inflammation)	1.00 ± 0.51	0.53 ± 0.19	0.36 ± 0.11	0.45 ± 0.32	0.64 ± 0.06	0.86 ± 0.86	n.s.	n.s.	n.s.
*Klotho* (FGF21 receptor)	1.00 ± 0.34	1.05 ± 0.50	1.23 ± 0.63	1.49 ± 0.68	0.50 ± 0.17	0.68 ± 0.31	n.s.	**	n.s.
*Leptin* (Energy metabolism)	1.00 ± 0.61	0.56 ± 0.39	0.55 ± 0.28	0.58 ± 0.36	1.89 ± 0.92	1.38 ± 0.66	n.s.	***	n.s.
*Prdm16* (Browning of WAT)	1.00 ± 0.29	0.88 ± 0.22	0.64 ± 0.30	0.54 ± 0.25	0.53 ± 0.11	1.08 ± 0.82	n.s.	n.s.	n.s.
*Pgc1α* (Browning of WAT)	1.00 ± 0.67	1.27 ± 0.68	1.20 ± 0.30	0.81 ± 0.46	0.29 ± 0.19	0.60 ± 0.28	n.s.	**	n.s.
*Pparγ* (Browning of WAT)	1.00 ± 0.51	1.23 ± 0.54	1.07 ± 0.48	0.84 ± 0.48	0.68 ± 0.23	1.02 ± 0.46	n.s.	n.s.	n.s.
*Ucp1* (Thermogenesis)	1.00 ± 0.83 ^d^	0.27 ± 0.14 ^d^	2.94 ± 1.46 ^d^	10.70 ± 9.16 ^abcef^	0.76 ± 0.15^d^	1.06 ± 1.44 ^d^	n.s.	***	*
*Cidea* (Thermogenesis)	1.00 ± 0.30	0.81 ± 0.25	1.32 ± 1.49	1.95 ± 3.09	0.32 ± 0.16	0.34 ± 0.18	n.s.	n.s.	n.s.
*Pparα* (Browning of WAT)	1.00 ± 0.66	0.75 ± 0.13	0.79 ± 0.39	0.62 ± 0.31	0.37 ± 0.17	0.68 ± 0.48	n.s.	n.s.	n.s.

WAT, white adipose tissue. Data (fold changes) are presented as the mean ± SD. *n* = 4–8 for each group. Con, Con+T, LPHF, LPHF+T, HPHF, and HPHF+T represent chow diet, chow diet plus training, low-protein high-fat diet, low-protein high-fat diet plus training, high-protein high-fat diet, and high-protein high-fat diet plus training, respectively. Feed, training, and interaction indicate a dominant effect of diet, treadmill exercise, and the interactive effect between feed and training, respectively. n.s., no significance was observed. * *p* < 0.05, ** *p* < 0.01, and *** *p* < 0.001. Significance in the differences between means was determined by Tukey’s post hoc test when ANOVA revealed a significant effect from the interaction of diet and training. ^a–f^ Significantly different from the Con, Con+T, LPHF, LPHF+T, HPHF, and HPHF+T groups, respectively.

**Table 6 metabolites-11-00301-t006:** Effects of LPHF, HPHF and/or training on BAT gene expression.

Gene	Con Groups		LPHF Groups		HPHF Groups		Significance		
	Con	Con+T	LPHF	LPHF+T	HPHF	HPHF+T	Training	Feed	Interaction
*Cidea* (Thermogenesis)	1.00 ± 0.15	1.20 ± 0.07	1.09 ± 0.53	1.09 ± 0.37	1.49 ± 0.36	0.96 ± 0.19	n.s.	n.s.	*
*Il-6* (Inflammation)	1.00 ± 0.51	1.45 ± 0.55	1.27 ± 0.63	1.55 ± 1.07	1.20 ± 0.91	0.83 ± 0.47	n.s.	n.s.	n.s.
*Prdm16* (Thermogenesis)	1.00 ± 0.28	1.15 ± 0.34	1.33 ± 0.31	1.18 ± 0.36	1.36 ± 0.36	0.94 ± 0.28	n.s.	n.s.	n.s.
*Ucp1* (Thermogenesis)	1.00 ± 0.24 ^ef^	1.60 ± 0.55 ^e^	1.36 ± 0.45 ^ef^	1.85 ± 0.30 ^e^	3.46 ± 0.95 ^abcd^	2.52 ± 0.66 ^ac^	n.s.	***	**
*Il-10* (Inflammation)	1.00 ± 1.17	0.37 ± 0.18	0.18 ± 0.06	0.40 ± 0.19	0.33 ± 0.24	0.30 ± 0.17	n.s.	n.s.	n.s.
*Cs* (Mitochondrial respiration)	1.00 ± 0.40	0.82 ± 0.25	0.86 ± 0.42	0.98 ± 0.43	0.77 ± 015	0.87 ± 0.44	n.s.	n.s.	n.s.

BAT, brown adipose tissue. Data (fold changes) are presented as the mean ± SD. *n* = 4–8 for each group. Con, Con+T, LPHF, LPHF+T, HPHF, and HPHF+T represent chow diet, chow diet plus training, low-protein high-fat diet, low-protein high-fat diet plus training, high-protein high-fat diet, and high-protein high-fat diet plus training, respectively. Feed, training, and interaction indicate a dominant effect of diet, treadmill exercise, and the interactive effect between feed and training, respectively. n.s., no significance was observed. Significance of differences between means was determined by Tukey’s post hoc test when ANOVA revealed a significant effect from the interaction of diet and training. * *p* < 0.05, ** *p* < 0.01, and *** *p* < 0.001. ^a–f^ Significantly different from the Con, Con+T, LPHF, LPHF+T, HPHF, and HPHF+T groups, respectively.

**Table 7 metabolites-11-00301-t007:** Effects of LPHF, HPHF, and/or training on gastrocnemius muscle gene expression.

Gene	Con Groups		LPHF Groups		HPHF Groups		Significance		
	Con	Con+T	LPHF	LPHF+T	HPHF	HPHF+T	Training	Feed	Interaction
*Acat1* (Ketone body metabolism)	1.00 ± 0.47	1.05 ± 0.44	1.05 ± 0.44	0.67 ± 0.33	1.00 ± 0.31	1.04 ± 0.50	n.s.	n.s.	n.s.
*Cd36* (Fatty acid transportation)	1.00 ± 0.28	0.99 ± 0.29	0.65 ± 0.15	0.83 ± 0.37	0.95 ± 0.37	1.40 ± 0.82	n.s.	n.s.	n.s.
*Cox4* (Mitochondrial respiration)	1.00 ± 0.30	0.97 ± 0.28	0.68 ± 0.26	0.66 ± 0.27	0.87 ± 0.39	1.26 ± 0.59	n.s.	*	n.s.
*Cs* (Mitochondrial respiration)	1.00 ± 0.23	1.20 ± 0.43	0.75 ± 0.38	0.76 ± 0.33	0.94 ± 0.42	1.33 ± 0.70	n.s.	n.s.	n.s.
*Cytochrome c* (Mitochondrial respiration)	1.00 ± 0.28	1.13 ± 0.44	0.87 ± 0.34	0.74 ± 0.29	1.13 ± 0.85	1.44 ± 0.69	n.s.	n.s.	n.s.
*Gp* (Glycogen phosphatase)	1.00 ± 0.25	1.21 ± 0.69	1.28 ± 0.34	0.31 ± 0.14	0.83 ± 0.32	1.13 ± 0.80	n.s.	n.s.	*
*Gs* (Glycogen synthase)	1.00 ± 0.40	0.84 ± 0.40	1.03 ± 0.37	0.98 ± 0.51	0.44 ± 0.16	0.79 ± 0.62	n.s.	n.s.	n.s.
*Hadh* (Fatty acid oxidation)	1.00 ± 0.27	0.96 ± 0.37	0.80 ± 0.41	0.64 ± 0.22	1.16 ± 0.64	1.28 ± 0.56	n.s.	*	n.s.
*Hbdh* (Ketone body metabolism)	1.00 ± 0.45	0.95 ± 0.44	0.44 ± 0.30	0.62 ± 0.33	1.04 ± 0.48	1.53 ± 0.67	n.s.	**	n.s.
*Mct1* (Ketone body transportation)	1.00 ± 0.22	0.93 ± 0.43	0.98 ± 0.54	0.62 ± 0.22	1.83 ± 0.94	1.53 ± 0.54	n.s.	**	n.s.
*Oxct1* (Ketone body metabolism)	1.00 ± 1.10	0.53 ± 0.30	0.33 ± 0.30	0.37 ± 0.31	0.61 ± 0.27	0.57 ± 0.24	n.s.	n.s.	n.s.
*Sirt1* (Mitochondrial respiration)	1.00 ± 0.39	0.79 ± 0.18	1.11 ± 0.30	0.82 ± 0.28	0.78 ± 0.20	0.63 ± 0.09	*	n.s.	n.s.
*Tfam* (Mitochondrial respiration)	1.00 ± 0.26	1.09 ± 0.42	0.96 ± 0.34	0.70 ± 0.22	1.18 ± 0.39	1.33 ± 0.69	n.s.	n.s.	n.s.
*Pparγ* (Glucose metabolism)	1.00 ± 0.25	1.11 ± 0.51	1.31 ± 1.13	0.76 ± 0.31	1.36 ± 0.66	1.35 ± 0.70	n.s.	n.s.	n.s.

Data (fold changes) are presented as the mean ± SD. *n* = 4–8 for each group. Con, Con+T, LPHF, LPHF+T, HPHF, and HPHF+T represent chow diet, chow diet plus training, low-protein high-fat diet, low-protein high-fat diet plus training, high-protein high-fat diet, and high-protein high-fat diet plus training, respectively. Feed, training, and interaction indicate a dominant effect of diet, treadmill exercise, and the interactive effect between feed and training, respectively. n.s., no significance was observed. * *p* < 0.05, and ** *p* < 0.01. Significance of differences between means was determined by Tukey’s post hoc test when ANOVA revealed a significant effect from interaction of diet and training.

**Table 8 metabolites-11-00301-t008:** Effects of LPHF, HPHF, and/or training on soleus muscle gene expression.

Gene	Con Groups		LPHF Groups		HPHF Groups		Significance		
	Con	Con+T	LPHF	LPHF+T	HPHF	HPHF+T	Training	Feed	Interaction
*Acat1* (Ketone body metabolism)	1.00 ± 0.35	0.51 ± 0.16	0.79 ± 0.19	0.53 ± 0.19	1.69 ± 0.74	2.32 ± 1.31	n.s.	***	n.s.
*Cox4* (Mitochondrial respiration)	1.00 ± 0.35	1.48 ± 0.20	0.96 ± 0.40	1.20 ± 0.46	1.71 ± 0.24	2.63 ± 1.00	*	***	n.s.
*Cpt1α* (Fatty acid oxidation)	1.00 ± 0.33	0.62 ± 0.17	1.05 ± 0.51	0.84 ± 0.31	3.17 ± 1.06	2.76 ± 1.22	n.s.	***	n.s.
*Hbdh* (Ketone body metabolism)	1.00 ± 0.51	0.70 ± 0.25	0.56 ± 0.12	0.48 ± 0.15	1.47 ± 0.57	1.79 ± 0.92	n.s.	**	n.s.
*Hk2* (Glycolysis)	1.00 ± 0.42	1.09 ± 0.13	0.70 ± 0.28	0.85 ± 0.37	2.11 ± 0.77	2.21 ± 1.04	n.s.	***	n.s.
*Il-6* (Fatty acid oxidation)	1.00 ± 0.57	1.05 ± 0.57	0.70 ± 0.51	0.80 ± 0.40	1.62 ± 1.10	1.37 ± 1.09	n.s.	n.s.	n.s.
*Mcad* (Fatty acid oxidation)	1.00 ± 0.64	0.55 ± 0.31	0.46 ± 0.54	0.81 ± 0.20	2.75 ± 2.88	3.49 ± 2.37	n.s.	**	n.s.
*Mct1* (Ketone body transportation)	1.00 ± 0.85	1.32 ± 0.52	1.92 ± 1.13	1.50 ± 0.66	5.51 ± 3.11	5.25 ± 1.69	n.s.	***	n.s.
*Tfam* (Mitochondrial respiration)	1.00 ± 0.53	0.83 ± 0.26	0.88 ± 0.31	1.01 ± 0.26	1.40 ± 0.43	1.94 ± 0.85	n.s.	**	n.s.
*Pparγ* (Glucose metabolism)	1.00 ± 0.50	0.33 ± 0.10	0.77 ± 0.47	0.50 ± 0.26	1.72 ± 0.62	2.23 ± 1.33	n.s.	***	n.s.
*Pgc1α* (Fatty acid mobilization)	1.00 ± 0.47	0.40 ± 0.09 ^f^	0.55 ± 0.48 ^f^	0.44 ± 0.23 ^f^	1.07 ± 0.35	1.79 ± 0.78 ^bcd^	n.s.	***	*
*Il-10* (Inflammation)	1.00 ± 0.93	0.42 ± 0.34	0.26 ± 0.28	0.27 ± 0.16	2.06 ± 2.24	4.17 ± 5.68	n.s.	*	n.s.
*Pepck* (Glycolysis)	1.00 ± 0.79	2.31 ± 2.02	6.74 ± 6.59	5.35 ± 6.05	3.39 ± 0.91	3.53 ± 1.67	n.s.	n.s.	n.s.
*Acc2* (Fatty acid synthesis)	1.00 ± 0.23	0.61 ± 0.21	0.43 ± 0.14	0.77 ± 0.43	1.46 ± 0.57	1.62 ± 0.82	n.s.	***	n.s.

Data (fold changes) are presented as the mean ± SD. n = 4–8 for each group. Con, Con+T, LPHF, LPHF+T, HPHF, and HPHF+T represent chow diet, chow diet plus training, low-protein high-fat diet, low-protein high-fat diet plus training, high-protein high-fat diet, and high-protein high-fat diet plus training, respectively. Feed, training, and interaction indicate a dominant effect of diet, treadmill exercise, and the interactive effect between feed and training, respectively. n.s., no significance was observed. * *p* < 0.05, ** *p* < 0.01, and *** *p* < 0.001. Significance of differences between means was determined by Tukey’s post hoc test when ANOVA revealed a significant effect from the interaction of diet and training. ^b–d^ Significantly different from the Con+T, LPHF, and LPHF+T groups, respectively. ^f^ Significantly different from the HPHF+T group.

**Table 9 metabolites-11-00301-t009:** Effects of LPHF, HPHF, and/or training on liver gene expression.

Gene	Con Groups		LPHF Groups		HPHF Groups		Significance		
	Con	Con+T	LPHF	LPHF+T	HPHF	HPHF+T	Training	Feed	Interaction
*F480* (Inflammation)	1.00 ± 0.26	0.95 ± 0.23	1.30 ± 0.76	0.72 ± 0.35	0.79 ± 0.29	0.72 ± 0.19	n.s.	n.s.	n.s.
*Hmgcs2* (Ketogenesis)	1.00 ± 0.56	0.65 ± 0.59	3.71 ± 2.30	2.73 ± 0.78	1.33 ± 0.99	1.26 ± 0.50	n.s.	***	n.s.
*Il-6* (Inflammation)	1.00 ± 0.57	0.70 ± 0.22	2.32 ± 3.37	0.31 ± 0.19	0.65 ± 0.41	0.92 ± 1.18	n.s.	n.s.	n.s.
*Pparγ* (Glucose metabolism)	1.00 ± 0.44	0.55 ± 0.13	1.10 ± 0.45	0.98 ± 0.26	1.06 ± 0.45	1.34 ± 0.60	n.s.	*	n.s.
*Cxcl2* (Inflammation)	1.00 ± 0.47	1.07 ± 1.08	0.55 ± 0.32	0.42 ± 0.31	0.51 ± 0.52	1.12 ± 0.92	n.s.	n.s.	n.s.
*Atgl* (Fatty acid mobilization)	1.00 ± 0.41	0.89 ± 0.60	0.81 ± 0.48	0.32 ± 0.15	0.34 ± 0.17	0.46 ± 0.34	n.s.	**	n.s.
*Fgf21* (Energy metabolism)	1.00 ± 1.11	0.57 ± 0.70	5.06 ± 3.88	3.75 ± 3.43	0.65 ± 0.33	1.15 ± 0.97	n.s.	***	n.s.
*Klotho* (FGF21 receptor)	1.00 ± 0.17	0.86 ± 0.24	0.85 ± 0.20	0.86 ± 0.30	0.60 ± 0.16	0.70 ± 0.18	n.s.	**	n.s.
*Pparα* (Fatty acid oxidation)	1.00 ± 0.42	0.55 ± 0.27	0.32 ± 0.16	0.39 ± 0.08	0.81 ± 0.38	0.89 ± 0.69	n.s.	***	n.s.
*Pgc1α* (Fatty acid mobilization)	1.00 ± 0.22	1.65 ± 1.26	0.66 ± 0.22	0.43 ± 0.09	0.36 ± 0.09	0.58 ± 0.74	n.s.	***	n.s.

Data (fold changes) are presented as the mean ± SD. n = 4–8 for each group. Con, Con+T, LPHF, LPHF+T, HPHF, and HPHF+T represent chow diet, chow diet plus training, low-protein high-fat diet, low-protein high-fat diet plus training, high-protein high-fat diet, and high-protein high-fat diet plus training, respectively. Feed, training, and interaction indicate a dominant effect of diet, treadmill exercise, and the interactive effect between feed and training, respectively. n.s., no significance was observed. * *p* < 0.05, ** *p* < 0.01, and *** *p* < 0.001.

**Table 10 metabolites-11-00301-t010:** Diet contents (in energy).

Component	Con	LPHF	HPHF
Protein (%)	14.2	4.8	14.2
Fat (%)	10.1	60.0	60.0
-Corn oil	10.1	8.7	8.7
-Lard	0.0	36.1	36.1
-Cream	0.0	15.2	15.2
Carbohydrate (%)	75.7	35.1	25.8
Kcal/g	3.6	5.5	5.3

**Table 11 metabolites-11-00301-t011:** List of primers used in real-time PCR.

Gene	Forward	Reverse
*18s*	CGGCTACCACATCCAAGGA	AGCTGGAATTACCGCGGC
*Gp*	TGGCAGAAGTGGTGAACAATGAC	CCGTGGAGATCTGCTCCGATA
*Adiponectin*	AGAGTCGTTGACGTTATCTGCATA	GGGCTCTGTGCTGCTCCATCT
*Gs*	ACTGCTTGGGCGTTATCTCTGTG	ATGCCCGCTCCATGCGTA
*Atgl*	GAGCCCCGGGGTGGAACAAGAT	AAAAGGTGGTGGGCAGGAGTAAGG
*Hadh*	ACTACATCAAAATGGGCTCTCAG	AGCAGAAATGGAATGCGGACC
*Cd36*	TGGCCTTACTTGGGATTGG	CCAGTGTATATGTAGGCTCATCCA
*Hbdh*	AGTTTGGGGTCGAGGCTTTC	TGGTGGCCGCTATGAAGTTG
*F480*	CTTTGGCTATGGGCTTCCAGTC	GCAAGGAGGACAGAGTTTATCGTG
*Mct1*	GCCTGAGCAAGTCAAGCTAG	TCAGACCTCGGATCCAGTAC
*Il-6*	AACGATGATGCACTTGCAGA	TGGTACTCCAGAAGACCAGAGG
*Oxct1*	CCAAGGAAGTAAATGAAGATCTCCTA	ACGTGTATGTTACAAGAAATGGCTTACC
*Klotho*	TGTTCTGCTGCGAGCTGTTAC	TACCGGACTCACGTACTGTTT
*Tfam*	TTCCCAAGACTTCATTTCATTGTC	GATGATTCGGCTCAGGGAAA
*Leptin*	GCTTTGGTCCTATCTGTCTTATGTT	CAATGGTCTTGATGAGGGTTTT
*Cpt1α*	CCAGGCTACAGTGGGACATT	GAACTTGCCCATGTCCTTGT
*Prdm16*	CCACCAGCGAGGACTTCA	GGAGGACTCTCGTAGCTCGAA
*Mcad*	GCTCGTGAGCACATTGAAAA	CATTGTCCAAAAGCCAAACC
*Pgc1α*	GACTGGAGGAAGACTAAACGGCCA	GCCAGTCACAGGAGGCATCTTT
*Cxcl2*	CCAACCACCAGGCTACAGG	GCGTCACACTCAAGCTCTG
*Pparγ*	CCACCAGCGAGGACTTCAC	GGAGGACTCTCGTAGCTCGAA
*Ucp1*	TGGTTGGTTTTATTCGTGGT	AGGGTTTGTGGCTTCTTTTC
*Cidea*	TCCTCGGCTGTCTCAATG	GGCTGCTCTTCTGTATCG
*Fgf21*	CAAGACACTGAAGCCCACCT	CACCCAGGATTTGAATGACC
*Pparα*	GAACCGGAACAAATGCCAGT	CTTCAGGTAGGCTTCGTGGA
*Il-10*	GCTCTTACTGACTGGCATGAG	CGCAGCTCTAGGAGCATGTG
*Sirt1*	GCAACAGCATCTTGCCTGAT	GTGCTACTGGTCTCACTT
*Cs*	GCAGCCAAGAACTCATCCTG	TCTGGGCCTGCTCCTTAGGTA
*Acat1*	CCGAGACAACTACCCAAGGA	CACACACAGGACCAGGACAC
*Cox4*	TGGGAGTGTTGTGAAGAGTGA	GCAGTGAAGCCGATGAAGAAC
*Cytochrome c*	CACGCTTTACCCTTCGTTCT	CTCATTTCCCTGCCATTCTC
*Hk2*	CTGTCTACAAGAAACATCCCCATTT	CACCGCCGTCACCATAGC
*Pepck*	ACAGTCATCATCACCCAAGAGC	CATAGGGCGAGTCTGTCAGTTC
*Acc2*	GACGCCCGAGGATCTGAAG	GGGACAGGGACGTACTGATC
*Hmgcs* *2*	AACTGGTGCAGAAATCTCTAGC	GGTTGAATAGCTCAGAACTAGCC

*18s*, 18s ribosomal RNA; *Gp*, glycogen phosphatase; *Gs*, glycogen synthase; *Atgl*, adipose triglyceride lipase; *Hadh*, hydroxyacyl-coA dehydrogenase; *Cd36*, cluster of differentiation 36; *Hbdh*, 3-hydroxybutyrate dehydrogenase; *Mct1*, monocarboxylate transporter 1; *Il-6*, interleukin-6; *Oxct1*, 3-oxoacid CoA-transferase 1; *Tfam*, mitochondrial transcription factor A; *Cpt1α*, carnitine palmitoyltransferase 1 alpha; *Prdm16*, PR domain containing 16; *Mcad*, medium chain acyl-CoA dehydrogenase; *Pgc1α*, peroxisome proliferator activated receptor gamma coactivator-1 alpha; *Cxcl2*, chemokine (C-X-C motif) ligand 2; *Pparγ*, peroxisome proliferator activated receptor gamma; *Ucp1*, mitochondrial uncoupling protein 1; *Cidea*, cell death inducing DNA fragmentation factor-alpha like effector A; *Fgf21*, fibroblast growth factor 21; *Pparα*, peroxisome proliferator activated receptor alpha; *Il-10*, interleukin-10; *Sirt1*, sirtuin1; *Cs*, citrate synthase; *Acat1*, acetyl-coenzyme A acetyltransferase 1; *Cox4*, cytochrome c oxidase polypeptide IV; *Hk2*, hexokinase 2; *Pepck*, phosphoenol-pyruvate carboxykinase; *Acc2*, Acetyl-CoA carboxylase 2; *Hmgcs2*, hydroxy-methylglutaryl-CoA synthase 2.

## Data Availability

The data presented in this study are available on request from the corresponding authors.

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
