# Peer review of "A Low-Protein High-Fat Diet Leads to Loss of Body Weight and White Adipose Tissue Weight via Enhancing Energy Expenditure in Mice"

_metabolites, 2021, doi:10.3390/metabo11050301_

Round 1

Reviewer 1 Report

To authors

Minor points

1) The authors confirm with two types of experimental diets that the protein content in the diet has a crucial impact on the metabolism of fats in different organs. LPHF reduces the ability of fat metabolism in WAT, muscle and liver. Also, the combination of diet and training has no additive effect.

2) What type of fatty acid has been used in the experimental diets? Please indicate this at manuscript.

3) Table 11. List of primers used in RT-PCR. You include Ly6g (lymphocyte antigen 6 complex) in the table, but no results appear with this data. Please remove the Ly6g primer or include the Ly6g results/discussion in the article.

4) In figure 4, what could be the cause of the training reducing the amount of FGF21?

5) Table 4. Add the decimal (,) to the amylase value in group HPHF + T.

6) In general for all tables: please try to better facilitate (or clarify) in the texts if the significant differences are based on a dominant effect of diets, training, or both. Its interpretation is sometimes incongruous.

For example:

  • Lines 104-105: “There existed a significant difference in SKM (skeletal muscle) weight between LPHF and HPHF”. These significant differences are not reflected in table 1. It only determines differences based on diet and exercise, in that case, explain it better about the text.
  • Line 108: “…while it significantly increased SKM…”. It seems that it is not significant or does not reflect it in table 1.
  • Lines 201-204: “Both LPHF 200 and HPHF significantly increased the levels of T-CHO, L-CHO, and H-CHO in plasma, compared to the chow diet. Furthermore, the levels of plasma T-CHO, L-CHO, and H-CHO were higher in HPHF groups than those in LPHF groups. When comparing to the chow diet, LPHF significantly decreased the plasma albumin and BUN”. This significance is not detailed in Table 3 between groups.
  • Lines 257-258: “Plasma amylase levels were significantly elevated by HPHF when comparing to Con”. It does not specify its significance in table 4.
  • Lines 261-262: “Plasma LDH was also increased by training in Con groups, despite no significance was observed”. It should be specified that it would be with LPHF + T.
  • Lines 292-297: “The expression of relevant metabolism-associated genes in various tissues and organs were analyzed. As presented in Table 5, compared to chow diet, HPHF significantly downregulated the expression of Atgl (fatty acid mobilization) and Klotho (FGF21 receptor), while significantly upregulated Leptin (energy metabolism) and Pgc1α (Browning of WAT) in WAT. The expression levels of Adiponectin (anti-inflammation) were significantly lower in HPHF than that in Con”.

In general, try to specify these differences in the text, whether it is a function of diet or training. The reader can look for these differences and not find them in table 5 (unless it is specified that it is based on diet or training). On the other hand, the expression of Pgc1α seems the opposite, it does not increase the expression, but is the most decreased in the HPHF group compared to the control.

  • Lines 352-359: “…LPHF significantly reduced the expression levels of Cox4 (mitochondrial respiration), compared to that in the chow diet. ….”  For example, with Cox4, the meanings in table 7 between control and LPHF group are not clearly indicated, although it is understood that you refer to the significance "based on diet" as the dominant effect.
  • Lines 391-392: “When comparing to Con, LPHF significantly downregulated Il-10”. Indicate the significance in table 8, or preferably on the text.
  • Lines 407-409: “As presented in Table 9, LPHF significantly upregulated the expression levels of Hmgcs2 (ketogenesis) and Fgf21 (energy metabolism), and downregulated Pparα (fatty 408 acid oxidation) and Pgc1α (fatty acid mobilization) in liver, compared to chow diet”. It doesn't seem to be significant.

Lines 410-413: “When comparing HPHF to Con, the expression levels of Atgl (fatty acid mobilization), Klotho (FGF21 receptor), and Pgc1α were significantly downregulated. The expression levels of Pparγ (glucose metabolism) were significantly higher in HPHF+T than those in Con+T”. This significant difference is not indicated in Table 9. Please clarify in the text again that the greatest significance in the liver would then be due to alterations with diet, not with training. 

7) Lines 495-501: “LPHF increased AST and lipase, indicators of hepatic damage and pancreatic damage, respectively. Previous studies reported that protein deficiency might lead to unfavorable changes of hepatic function via lowering hepatic antioxidative activity [42], as well as undesirable alterations of pancreatic function by altering secretory synergism in islets [43]. Amylase was increased by HPHF. Amylase was used as another marker of pancreatic damage”. But have you verified that your models actually suffer from damage to those organs? In any case there is only a change in its physiology, but it does not present damage.

Major points

8) You indicate that the increase in the relative proportion of liver in the LPHF group could be attributed to the loss of body weight. Furthermore, in LPHF the energy intake does not seem lower than in HPHF and induces a greater appetite.

The animals on the two experimental diets had the same energy availability, but did you measure energy intake? Was an energy balance made with measurement of food intake and excretion?

9) You also raise the possibility of weight loss with LPHF due to increased energy expenditure instead of appetite suppression, attributed to the overexpression of Ucp1 in WAT (and Fgf21 in liver). Together with the energy use (expenditure) with LPHF, can the decrease in body weight and WAT weight in the LPHF group be conditioned by steatorrhea? Have you been able to check the presence of fatty stools and measure the amount of fat in the stool?

Furthermore, you indicate that there is no significant difference in CD36 expression (for fatty acid transport) in WAT. Do you not consider a greater role of the expression of CD36 on Ucp1?

10) What could be the reason that the expression of Ucp1 increases in WAT tissue with LPHF + T and, however, with BAT it occurs in HPHF + T?

Reviewer 2 Report

The manuscript with the title ”A Low Protein, High Fat Diet leads to loss of body weight and 2 white adipose tissue weight via enhancing energy expenditure 3 in Mice” by Rang et al investigates the impact of low-protein high fat diet on weight loss. The study is highly relevant and timely, given the recent public interest in diets such as low carb high fat diets and other variants of this. The low protein high fat diet studied here is certainly interesting, given recent alterations in food patterns in the population. The study is well conducted, designed and described. However, I am not convinced that the data presented support the use of this diet for weight loss, as concluded by the authors “Therefore, LPHF, however, without training, may be potential for body weight management”.

In the introduction the authors focus on energy expenditure. It would be nice to see some introduction also about protein restriction in combination with physical activity, a combination that is studied here and that is seemingly incompatible. What does previous studies tell about protein needs in relation to exercise and muscle physiology – this must have been taken into account when designing the present study?

The authors observe reduced WAT, but increased liver and muscle mass in the LPLF diet. However, the increase in muscle mass does not translate into an increased strength. Also, indications of increased muscle damage may be difficult to understand in this scenario. The increased liver mass is worrying. The authors discuss that these findings may be affected by the use of relative tissue mass. It would be interesting to see absolute weights, and, from a metabolic point of view, also fat content in muscle and liver. Especially, liver fat and histology are necessary to asses to be able to draw the conclusion that “LPHF might be suitable for individuals with normal weight to maintain a lower weight”. The author may also wish to rephrase this conclusion (row 434), as weight loss in normal weight individuals are difficult to motivate.

The authors also discuss (first section page 16) potential implications of satiety and caloric intake. Did the authors measure food intake? This would be necessary in able to assess whether the results are due to increased basal metabolic rate or just a reduction in caloric intake. This data should be included or otherwise stated as a severe limitation of the present study.

In humans, increased training is often associated with increased food intake (and therefore marginal impact on body weight in a free-fed situation). This seem not to be the case in the presently used mouse model, requiring a deeper discussion on the applicability of the findings with respect to human weight loss interventions.

When during the day were mice trained? Could this have some implications on the results, i.e. training when they are normally inactive or active and eating?

Same with the fasting procedure. Where they fasted daytime or nighttime? Depending on which, is the procedure be considered as fasting or starvation?

In relation to this, indications of muscle and tissue damage may need further examination. For instance, is it possible that lower glycogen levels in the LPHF diet makes these mice more sensitive to starvation? Do the observed changes depend on the length of the starvation period?

With respect to the ANOVA analyses, it is not clear in the statistical methods that an interaction is included in the model. Please clarify. Also, as there may be significant variation between mice. Did the authors consider a repeated measures design to analyze longitudinal data, e.g. weight-gain trajectories?

A total of 32 genes are assayed in this study. Hence, results may be impacted by multiple testing issues. Please adjust for this.

An overall reflection is that the authors describe the non-exercised mice as having a sendentary lifestyle. Is this really correct, i.e. are they “exercising” less than an ordinary lab mouse?

Overall, a lot of the mechanistic understanding is based on gene expression studies. I would like to see at least one of the hypotheses tested by an alternative assay that is more directly linked to the activity of highlighted enzymes/pathways. Conclusions as to if fatty acid or glucose oxidative metabolism is reduced or not are difficult to draw without knowing the activity of relevant metabolic pathways.

Round 2

Reviewer 1 Report

1) I congratulate the authors for the improvements made to the article and thank all the amendments that were requested.

2) I appreciate that the authors have clarified the principle of interaction and the significance of the impact by comparing the smallest or largest values of the main factors (feed or training).

3) It is also appreciated that the authors acknowledge the limitations of their study and propose improvements for future studies.

4) The inclusion of Figure S1 is valued to know the dietary intake through the changes of intake fodder weight. 

Author Response

Thank you for your precious suggestions, which help us to consider in-depth about our research and make our paper more meaningful for future studies.

Reviewer 2 Report

Congratulations on a well written manuscript. You have responded to all my comments. Just one thing remaining: I would still suggest a correction for multiple testing in the expression analysis. Then, you may still discuss changes that do not reach multiple testing adjusted significance, while stating that they differ at a nominal level. 

Author Response

Thank you for your precious suggestion. In the Discussion, we have considered the gene expression which changed obviously while did not exhibit significance. For example, Pepck (glycolysis) in soleus muscle (in Table 8 in Line 331) increased dramatically (over 3 times in means) while no significance was observed. We discussed this change in Line 519-523. As to the other gene expression with similar changes such as Il-6 in WAT (in Table 5 in Line 271), Il-10 in BAT (in Table 6 in Line 286), and Cd36 in gastrocnemius muscle (in Table 7 in Line 309), considering the standard deviation (SD) are relatively large and the difference of means does not exceed 3 times, it seems not to be convinced to make a speculation or regard as a finding. In addition, in order to make it clear that we conducted multiple testing for each data with significant effect of interaction of diet and training, we added note “Significance of differences between means was determined by Tukey’s post hoc test when ANOVA revealed a significant effect from interaction of diet and training” in Line 102-103, 125-126, 165-166, 189-190, 221-222, 230-231, 251-252, 276-277, 291-292, 314-315, 336-337, as well as in the supplementary data. This note was marked in yellow in the supplementary data.